# Multiomics Reveals Symbionts, Pathogens, and Tissue-Specific Microbiome of Blacklegged Ticks (*Ixodes scapularis*) from a Lyme Disease Hot Spot in Southeastern Ontario, Canada

Amber R. Paulson,[a*] Stephen C. Lougheed,[a] David Huang,[a] Robert I. Colautti[a]

aDepartment of Biology, Queen's University, Kingston, Ontario, Canada

**ABSTRACT** Ticks in the family *Ixodidae* are important vectors of zoonoses, including Lyme disease (LD), which is caused by spirochete bacteria from the *Borreliella* (*Borrelia*) *burgdorferi sensu lato* complex. The blacklegged tick (*Ixodes scapularis*) continues to expand across Canada, creating hot spots of elevated LD risk at the leading edge of its expanding range. Current efforts to understand the risk of pathogen transmission associated with *I. scapularis* in Canada focus primarily on targeted screens, while natural variation in the tick microbiome remains poorly understood. Using multiomics consisting of 16S metabarcoding and ribosome-depleted, whole-shotgun RNA transcriptome sequencing, we examined the microbial communities associated with adult *I. scapularis* (*n* = 32), sampled from four tissue types (whole tick, salivary glands, midgut, and viscera) and three geographical locations within a LD hot spot near Kingston, Ontario, Canada. The communities consisted of both endosymbiotic and known or potentially pathogenic microbes, including RNA viruses, bacteria, and a *Babesia* sp. intracellular parasite. We show that $\beta$-diversity is significantly higher between the bacterial communities of individual tick salivary glands and midguts than that of whole ticks. Linear discriminant analysis effect size (LEfSe) determined that the three potentially pathogenic bacteria detected by V4 16S rRNA sequencing also differed among dissected tissues only, including a *Borrelia* strain from the *B. burgdorferi sensu lato* complex, *Borrelia miyamotoi*, and *Anaplasma phagocytophilum*. Importantly, we find coinfection of *I. scapularis* by multiple microbes, in contrast to diagnostic protocols for LD, which typically focus on infection from a single pathogen of interest (*B. burgdorferi sensu stricto*).

**IMPORTANCE** As a vector of human health concern, blacklegged ticks (*Ixodes scapularis*) transmit pathogens that cause tick-borne diseases (TBDs), including Lyme disease (LD). Several hot spots of elevated LD risk have emerged across Canada as *I. scapularis* expands its range. Focusing on a hot spot in southeastern Ontario, we used high-throughput sequencing to characterize the microbiome of whole ticks and dissected salivary glands and midguts. Compared with whole ticks, salivary glands and midguts were more diverse and associated with distinct bacterial communities that are less dominated by *Rickettsia* endosymbiont bacteria and are enriched for pathogenic bacteria, including a *B. burgdorferi sensu lato*-associated *Borrelia* sp., *Borrelia miyamotoi*, and *Anaplasma phagocytophilum*. We also found evidence of coinfection of *I. scapularis* by multiple pathogens. Overall, our study highlights the challenges and opportunities associated with the surveillance of the microbiome of *I. scapularis* for pathogen detection using metabarcoding and metatranscriptome approaches.

**KEYWORDS** *Bunyavirales*, Lyme disease, tick-borne disease, anaplasmosis, babesiosis, blacklegged tick, endosymbiont, tick-borne relapsing fever, vector, zoonosis

Address correspondence to Amber R. Paulson, Amber.Rose.Paulson@gmail.com, or Robert I. Colautti, Robert.Colautti@queensu.ca.

*Present address: Amber. R. Paulson, Environmental Assessment Office, BC Ministry of Environment and Climate Change Strategy, Victoria, British Columbia, Canada.

The authors declare no conflict of interest.

One-fifth of emerging human infectious diseases are transmitted by arthropod vectors, of which 40% are transmitted by Ixodidae ticks, representing an increasingly large threat to human health and welfare (1, 2). Ticks in the genus *Ixodes* are important

vectors of human disease, including LD, tick-borne relapsing fever, anaplasmosis, babesiosis, ehrlichiosis, tick-borne encephalitis, and Powassan virus (3–6). In Eastern North America, *I. scapularis* is the primary vector of LD, contributing to the more than 300,000 estimated cases in the United States each year (7, 8). In addition to *Borrelia burgdorferi sensu stricto*, *I. scapularis* is also a vector for the broader *B. burgdorferi sensu lato* complex of spirochete pathogens, including *Borrelia mayonii*, *Borrelia kurtenbachii*, *Borrelia bissettiae*, and *Borrelia andersonii* (9–14).

In addition to *B. burgdorferi sensu lato* spirochetes, a variety of other co-occurring pathogens have been detected in *I. scapularis*, including *Anaplasma phagocytophilum*, *Borrelia miyamotoi*, *Babesia microti*, *Ehrlichia muris eauclarensis*, and Powassan virus (15). The growing list of pathogens harbored by *I. scapularis* increases the potential for polymicrobial infections, which can complicate diagnosis and result in increased disease severity (16, 17). While advances in high-throughput sequencing (HTS) have enabled broader detection and monitoring of co-occurring pathogens and microbes harbored by ticks, this technique is not often used for tick monitoring in Canada. Moreover, the high sensitivity of HTS holds potential for characterizing microbial communities across different tissues of the same tick.

The gut, salivary gland, and other tissues represent different environments that may select for different bacterial communities. Indeed, experimental dysbiosis of the tick microbiome has revealed complex interactions among the bacterial community that can influence pathogen colonization in the midgut (18–21). It has also been shown that *I. scapularis* produces antimicrobial factors that can modulate microbe colonization (21–23). This experimental evidence shows how tissues represent different microbial environments, but it is not clear how the variation is affected among tissues from ticks collected *in situ*. Given the complex interactions that occur in the midgut and the salivary glands of *I. scapularis*, further screening of these tissues from field-collected ticks using HTS can provide a new dimension for understanding the interplay between pathogen and tick microbiome and their implications for animal and human health.

In addition to immunological factors, the effects of ecological interactions within the tick microbiome remain poorly understood in *I. scapularis*. Recent studies in *I. scapularis* have revealed maternally transmitted endosymbiotic bacteria, *Rickettsia buchneri*, in relatively high abundances in female *I. scapularis* compared with those in males and at higher levels following engorgement (24–27). A negative co-occurrence between *B. burgdorferi* and *R. buchneri* was found among male *I. scapularis* (28), while another study found a potentially negative correlation between *B. burgdorferi* and "environmental bacteria," including *Bacillus* sp., *Pseudomonas* sp., or uncharacterized *Enterobacteriaceae* (29), implying the potential for competition or some other form of interaction to occur between different bacterial associates. Experimental work has shown that *R. buchneri* inhibits the growth of other pathogenic *Rickettsiaceae* in tick cell lines (30), which may be related to the expression of cryptic genes for interbacterial antagonism (31). Such findings suggest that endosymbionts may influence TBD risk, but to our knowledge, co-occurrence between *R. buchneri* and microbial pathogens has not been assessed in wild populations of *I. scapularis* at the expanding range in Canada.

Although most are not known as human pathogens, there are reported cases of tick endosymbiont transmission to humans (32–35). Indeed, many tick endosymbionts are closely related to human pathogens, with examples of evolutionary transitions between pathogenic and symbiotic lifestyles occurring within the spotted fever group of *Rickettsia* (36, 37), *Coxiella* spp. (38), and the *Francisella*/FLE-like group (39). These bidirectional transitions represent a symbiont-pathogen continuum, further emphasizing the value of HTS surveillance of tick microbiome communities to complement more targeted monitoring initiatives.

While bacteria are commonly studied, ticks also harbor archaea, viruses, and other eukaryotic parasites (16, 40, 41), and their disease risks are mostly unknown. To complement sequencing of 16S rRNA genes for bacteria, rRNA-depleted metatranscriptome

sequencing offers a complementary approach to identify bacterial and nonbacterial associates and to capture functional genes expressed within the microbiome. Metatranscriptome studies in *I. scapularis* have revealed cross-kingdom interactions that occur within the microbiome of the tick and may affect TBD risk. For example, nonrandom co-occurrences have been detected between *Babesia microti* and black-legged tick phlebovirus (BTPV), *B. burgdorferi* and BTPV (42), and South Bay virus (SBV) and *B. burgdorferi* (43). More generally, metatranscriptome investigations of tick microbial gene expression have revealed the unprecedented diversity of RNA viruses (42–44), including novel pathogens (45, 46).

Recent geographic range expansion of *I. scapularis* is occurring across the United States (47) and into Canada (48–51). Northward expansion of *I. scapularis* ticks into southern Canada is patchy, resulting in regional LD hot spots and geographic variation in microbial communities (52–55). Microbiome and transcriptome data from *Ixodes* ticks are rare in Canada but could provide insight into how range expansion shapes microbial communities and the epidemiology of TBDs as part of a transdisciplinary approach advocated by Talbot et al. (56). Specifically, metabarcoding and metatranscriptome data from ticks collected in field surveys provide "ecological evidence" and "molecular evidence" to inform epidemiological models and targets for diagnostic tests. The goal of our study was to understand how tick biology and environment contribute to microbiome composition and pathogen prevalence in natural tick populations, using complementary approaches. Here, we present results of HTS-based surveillance of *I. scapularis* microbiomes in an LD hot spot in Canada, using both targeted sequencing of the V4 region of the 16S rRNA gene and rRNA-depleted, whole shotgun RNA metatranscriptome sequencing. We examined tissue-specific differences in bacterial microbiomes between the salivary gland, midgut, and other internal viscera, as well as whole-tick extractions, from three distinct sites representing different habitats and land-use histories. Our study demonstrates how HTS methods applied to individual tick dissections can help identify changing health risks associated with range expansion by an arthropod vector of increasing public health concern.

## RESULTS

**Bacterial community inferred from V4 by 16S rRNA gene sequencing.** The core bacterial community associated with *I. scapularis* was represented by 25 dominant amplicon sequencing variants (ASVs), each with a minimum relative abundance of greater than 0.1% (Fig. 1; see Table S1 in the supplemental material). Within the core, only *A. phagocytophilum* (ASV8) and *B. miyamotoi* (ASV2) were identified at the species level by the DADA2 pipeline. We identified three primary ASVs of concern (AoCs) from the core microbiome, which included *Borrelia* sp. (ASV3), *B. miyamotoi*, and *A. phagocytophilum*. Among these AoCs, the potentially LD-causing spirochete *Borrelia* sp. was the most widely detected one across all tissue types and sampling sites in this study, ranging from trace levels up to 98% relative abundance. We also detected the relapsing fever spirochete *B. miyamotoi* at >99% relative abundance in the salivary gland and midgut of a single tick collected at the Lemoine Point (LP) site.

Aside from the three AoCs detected in the core microbiome, another major feature of the *I. scapularis* bacterial community was the widespread detection of *Rickettsia* sp. (ASV1), which was common across all three sites from whole-tick and dissected tissue samples. A total of six ASVs were identified as *Rickettsia* sp. (see Table S2 in the supplemental material) with the sequences ranging in length from 254 to 258 bp and sharing 93.8 to 99.6% sequence similarity based on multiple sequence alignment (CLUSTAL O [1.2.4]) (data not shown). Of these ASVs, ASV1 contributed >99.95% of all *Rickettsia*-associated sequences captured in this study, which shares 100% homology to the V4 region of the 16S rRNA of *R. buchneri* (endosymbiont of *I. scapularis*). Also notable, *Williamsia* sp. (ASV16) and *Mycobacterium* sp. (ASV11), *Sphingomonas* sp. (ASV4), and *Pseudomonas* sp. (ASV5) were detected in the core microbiome at two or more of the sampling sites (Fig. 1).

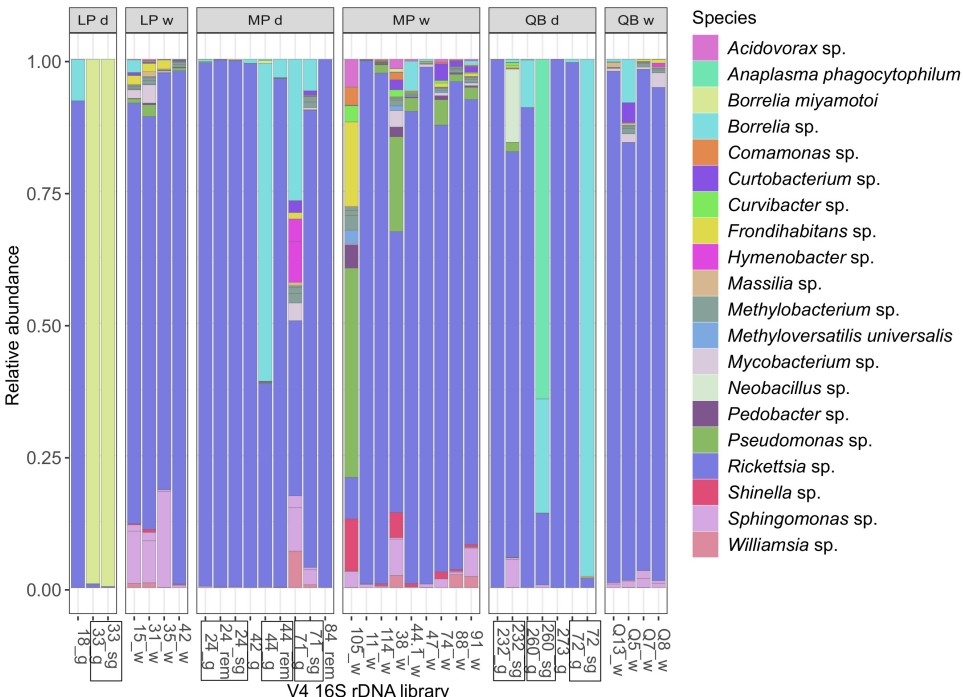

**FIG 1** Relative abundance of amplicon sequence variants (ASVs) of the V4 16S rRNA region detected in the core microbiome of *I. scapularis*. Core bacteria are defined as ASVs with a minimum relative abundance of 0.1% in at least one sequencing library. Libraries originating from different tissues of the same individual are denoted by black rectangles around the sample identifiers on the *x* axis. Sample types are salivary gland (sg), gut (g), remaining internal viscera (rem), whole (w), or dissected (d) in the library identifiers/facet titles. Samples were collected from the following three locations in eastern Ontario: Lemoine Point (LP), Murphy's Point (MP), and Queen's University Biological Station (QB) (see facet titles).

We consider the detection of ASVs from the sequencing libraries to be reliable. The negative-control PCR and bead libraries contained markedly fewer sequences following our data processing but removed one salivary gland-based sample (42_sg) (see Fig. S1 in the supplemental material), noting that an initial inspection of ASVs detected among the three separate batch PCR replicate libraries determined the presence of several additional *Rickettsia*-associated ASVs almost exclusively from a single PCR batch, including the negative-control PCR (see Table S3 and Fig. S2 in the supplemental material). Since these results were unexpected, the entire batch replicate was removed from further analysis, while the two more consistent batch replicates were retained (noting that seven of the dissected samples had only a single PCR batch replicate).

We identified, 305 potentially contaminating ASVs using the "decontam" package, which were removed from the dataset (see Table S4 in the supplemental material). These sequences included taxa typically known from environmental samples (e.g., *Sediminibacterium*, *Pseudomonas*, *Paraburkholderia*, *Enhydrobacter*, and *Massilia*) or human sources (e.g., *Staphylococcus*, *Streptococcus*, *Acinetobacter*, *Cutibacterium*, *Chryseobacterium*, and *Corynebacterium*). Based on the asymptotes of rarefaction curves, bacterial communities associated with whole ticks generally contained more ASVs than those detected from the various dissected tissues (see Fig. S3 in the supplemental material). Overall, each library contained 1 to 139 ASVs, with an average of 54.

Sequences attributed to ASV7 were detected in 2 out of 28 of the tick samples, which were both from salivary gland samples from the Queen's University Biological Station (QB) site and could not be assigned beyond kingdom *Bacteria* in our pipeline (see Table S5 in the supplemental material). However, a BLAST search of the GenBank nucleotide database identified similarity (E value of $3e^{-106}$; 248/264 identities) to the apicoplast of the *Babesia* sp. Dunhaung (MH992225.1) and *Babesia* sp. Xinjiang complete genome (MH992224.1)

and the Kashi isolate (KX881914.1) (57). Phylogenetic analysis placed ASV7 within the Clade X *Babesia* "*sensu stricto*" (i.e., classical or true *Babesia*) (see Fig. S4 in the supplemental material). The *Babesia*-associated ASV7 was not included in further analysis.

Coinfection of ticks with multiple AoCs was identified in the salivary gland of a tick from the QB site (individual 260_sg). This sample was coinfected with *Borrelia* sp. and *A. phagocytophilum*, representing 22% and 64% relative abundance, respectively (Fig. 1). In contrast, the midgut of the same individual (260_g) contained 9% relative abundance of *Borrelia* sp. and trace levels (<1% relative abundance) of *A. phagocytophilum*. Moreover, the *Babesia* sp. classified as ASV7 (see above) was detected among other bacterial AoCs, either coinfecting the salivary gland with *Borrelia* sp. in sample 72_sg or present in a triple infection with both *Borrelia* sp. and *A. phagocytophilum* in 260_sg, which were both from a salivary gland collected at the QB site.

Using an additional phylogenetic analysis, the 258 bp V4 16S rRNA sequence from *A. phagocytophilum* was 100% identical to sequences from strain Ap-var-1, which was detected previously in Canada and distinct from the known pathogenic strain Ap-ha also detected in Canada (58) (accession HG916767.1) (Fig. S5). The sequence from *Borrelia* sp. matched 100% with the partial 16S rRNA sequence from *B. bissettiae* DN127 (255/255-bp identities; NR_114707.1) (59) and *B. burgdorferi sensu stricto* isolate 15-0797 (255/255 bp identities; MH781147.1) (see Fig. S6 in the supplemental material). The V4 16S rRNA sequence from *B. miyamotoi* matched 100% with the *B. miyamotoi* strain HT31 (NR_025861) with 257/257-bp identities, which is distinct from the *B. burgdorferi sensu lato* complex (Fig. S6).

We used linear discriminant analysis effect size (LEfSe) to explore the variation in ASV abundance among tissue types and sample sites. The abundance of two AoCs of concern, namely, *Borrelia* sp. and *A. phagocytophilum*, discriminated between dissected tick tissues and whole-tick samples, based on LDA scores (see Fig. S7A in the supplemental material). A third AoC, *B. miyamotoi*, discriminated between tissue samples from the LP site only (Fig. S7B). While only the abundance of the three AoCs discriminated among dissected tick tissues, LEfSe identified several core ASVs that discriminate among sites for the whole-tick samples (Fig. S7B). Specifically, *Methylobacterium* sp., *Mycobacterium* sp., *Curtobacterium* sp., *Massilia* sp., *Hymenobacter* sp., and order *Rhizobiales* were more common at the QB site; *Pseudomonas* sp. and *Shinella* were more common at the MP site; and *Sphingomonas* sp., *Frondihabitans* sp., and *Clavibacter* sp. were more common at the LP site.

Consistent with LEfSe, principal coordinate analysis (PCoA) of the V4 16S rRNA libraries using weighted UniFrac distances separated samples primarily by a high abundance of AoCs, which explain 89.5% of the total variation among the bacterial communities associated with *I. scapularis* (Fig. 2). Based on the PCoA with weighted UniFrac distances, some of the salivary glands and midguts have a distinct microbial community compared with the whole tick samples, which are defined by a high relative abundance of *Rickettsia* sp. clustered together (bottom-left corner in Fig. 2), whereas those with relatively high abundances of *Borrelia* sp., *B. miyamotoi*, *A. phagocytophilum*, or *Pseudomonas* sp. were separated from the main cluster of samples.

Using an unweighted UniFrac distance-based PCoA for the whole bacterial microbiome, we found that the communities differed across tissues compared with those of the whole-tick samples with the first two axes explaining 34.1% of the variation in the data (Fig. 3A). Generally, the unweighted UniFrac ordination separated the bacterial communities by tissue type, with whole-tick samples clustering more closely and away from dissected samples. Overall, we detected 409 ASVs from the ticks and tissues that were sampled, with 112 ASVs unique to whole ticks, 152 unique to dissected tissues, and 145 shared between them (Fig. 3B). Among the unique ASVs, strains of bacteria in the phylum *Verrucomicrobia* were more common in whole-body samples from the LP and MP sites, compared with strains in the phylum *Firmicutes*, which were more common among dissected tissues collected across all three sampling locations (see Fig. S8 in the supplemental material).

Differences in $\beta$-diversity between communities within individual female ticks sampled from whole or dissected tissues were evident. The average unweighted UniFrac

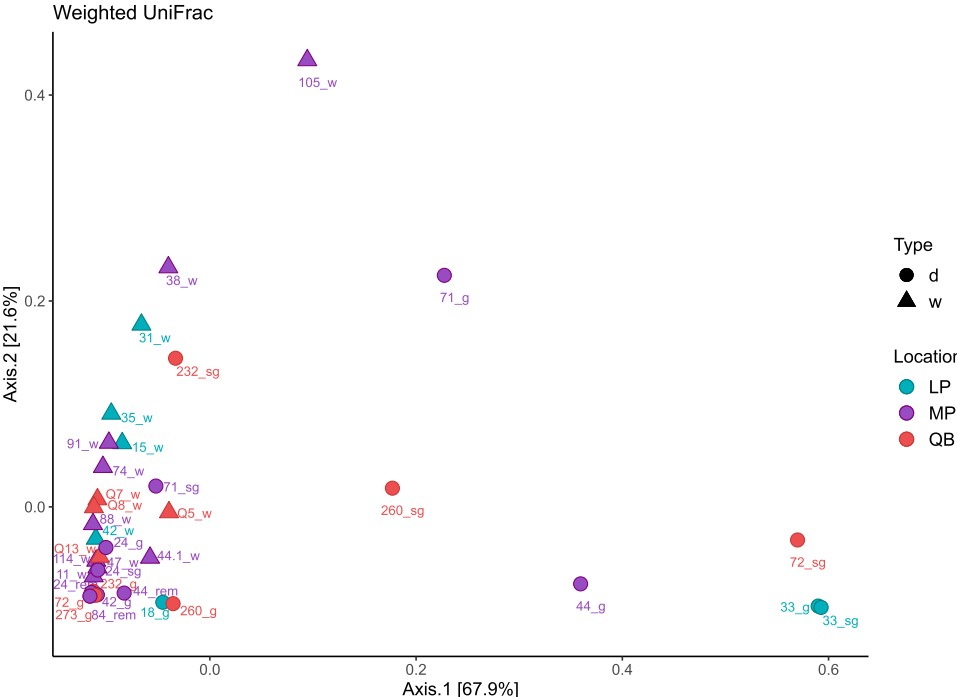

**FIG 2** Principal coordinate analysis (PCoA) of weighted UniFrac distances for ASVs detected from *I. scapularis* using V4 16S rRNA sequencing (all ASVs were agglomerated at the lowest taxonomic level; $n$ = 419). Sample types are indicated as salivary gland (sg), gut (g), internal viscera (rem), or whole (w) in library names. Samples were collected at three locations, as follows: Lemoine's Point (LP), Murphy's Point (MP), and Queen's University Biological Station (QB). Shapes indicate tissue type, namely, whole (w) or dissected (d).

distance of 0.512 (SD, 0.078) between whole tick libraries was significantly lower ($P <$ $2.22e^{-16}$) than the average distance of 0.707 (SD, 0.118) among libraries from dissected ticks (see Fig. S9 in the supplemental material). Furthermore, the average distance among the whole-tick libraries was also significantly lower ($P < 2.22e^{-16}$) than the average distance of 0.706 (SD, 0.104) measured from libraries between the two different groups. Using a permutation test for the homogeneity of multivariate dispersions, we also identified a significant difference in variance between whole and dissected sample types. The unweighted UniFrac PCoA was repeated using only the top 500 most relatively abundant ASVs, which was consistent with the previous findings that used ASVs agglomerated at the lowest taxonomic level (see Fig. S10A in the supplemental material), but PCoA inverted axis 2 and showed slightly more distinction between dissected and whole-tick samples (Fig. S10B).

**Whole shotgun RNA sequencing.** Following the removal of transcripts associated with unwanted phyla (see Materials and Methods), we further analyzed the remaining 402 transcripts, of which endosymbiont *R. buchneri* comprised over 75% (Table 1). Single-stranded negative-sense RNA viruses were common in the *I. scapularis* metatranscriptome (Fig. 4), including transcripts for N and L proteins from two different viruses in the family *Bunyaviridae*. Of these viruses, a South Bay virus (SBV) from the genus *Nairovirus* was found to be highly expressed in all four tick samples. We also identified transcripts from two variants of blacklegged tick phlebovirus (BLTPV) from the genus *Phlebovirus* of *Bunyaviridae* and from three different variants of the currently unclassified *Ixodes scapularis*-associated virus (ISAV) (Fig. 4). Finally, a partial N protein transcript that was 508 bp long shared sequence similarity with the Chimay rhabdovirus N protein by BLASTx (E value = $1.69e^{-17}$), but it was expressed at low levels relative to the other virus-associated transcripts.

Transcripts of the eukaryotic parasite *Babesia divergens* were identified in low abundance relative to the virus-associated sequences. Transcripts of prokaryotic origin were identified at intermediate abundance, including *R. buchneri* and *A. phagocytophilum*,

**A**

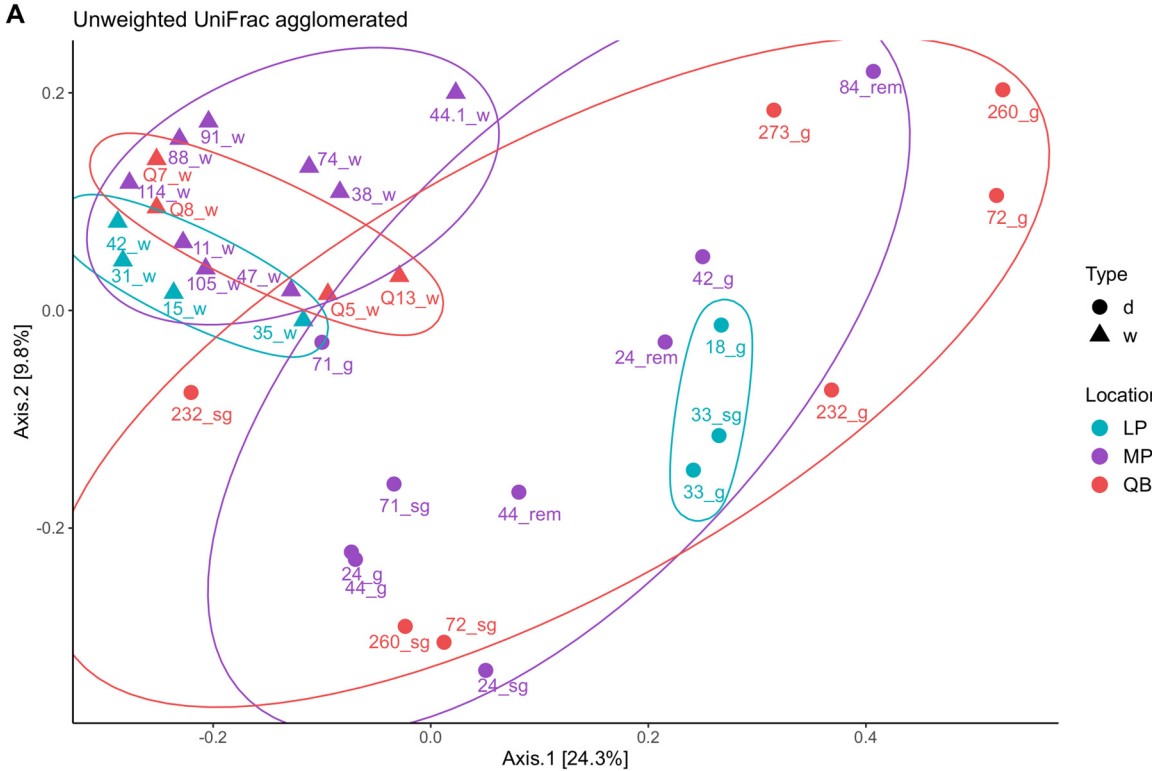

**B**

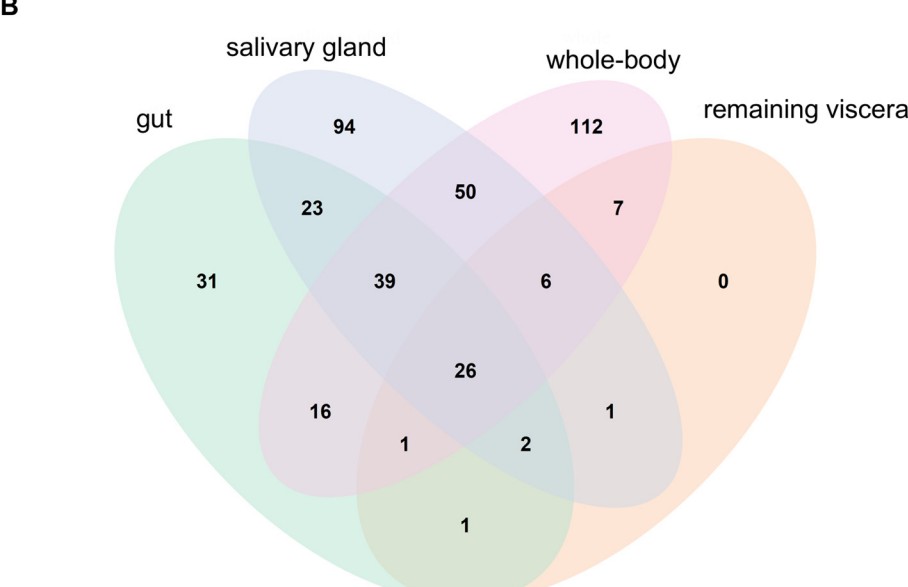

**FIG 3** (A) Principal coordinate analysis (PCoA) of unweighted UniFrac distances for ASVs. The tissue types are indicated in the library names as either dissected (d) salivary gland (sg), gut (g), and internal viscera (rem) or whole (w). Location abbreviations are as follows: Lemoine's Point (LP), Murphy's Point (MP), and Queen's University Biological Station (QB). (B) Venn diagram of ASVs detected using V4 16S rRNA sequencing from the whole *I. scapularis* or dissected salivary gland, midgut, and other remaining internal viscera.

while *Borrelia*-associated transcripts were captured at relatively lower abundances. Forty *R. buchneri* genes were expressed in *I. scapularis*, including genes encoding *bioB* (involved in biotin synthesis), outer membrane proteins, lipoproteins, peptidoglycan modification, autotransporters, porins, heat shock proteins, transposases, molecular chaperones, DNA methylation, ribosome-associated proteins, and other hypothetical genes of unknown function.

**TABLE 1** Proportion of taxonomic assignments in the *I. scapularis* metatranscriptome

| Super kingdom | Annotation | Proportion (%) |
|---|---|---|
| Eukaryota | Other nonarthropod | 10.70 |
| Prokaryota | *R. buchneri* | 75.12 |
| Prokaryota | *R. parkeri* | 0.50 |
| Prokaryota | *B. burgdorferi* | 1.24 |
| Prokaryota | *B. miyamotoi* | 0.25 |
| Prokaryota | *A. phagocytophilum* | 3.98 |
| Prokaryota | Other | 3.48 |
| Viruses | Viruses | 4.73 |

## DISCUSSION

Using a multiomics approach with HTS, we found evidence for endosymbiotic and pathogenic microorganisms of viral, eukaryotic, and prokaryotic origin associated with adult *I. scapularis* collected from a LD hot spot located in southeastern Ontario. Consistent with 16S rRNA-based sequencing studies of *I. scapularis* from this and other regions of eastern North America, the most common bacteria with the highest relative abundance detected in our study belong to the genus *Rickettsia* (29, 60–65). Also consistent with studies in other geographic regions, we found three primary AoCs, including *A. phagocytophilum*, a spirochete from the *B. burgdorferi sensu lato* complex (*Borrelia* sp.), and the non-*B. burgdorferi sensu lato* relapsing fever spirochete *B. miyamotoi*. Using whole shotgun RNA sequencing we found that *I. scapularis* was infected with these same three primary AoCs, as well *Babesia* sp. and *R. buchneri*, although few pathogen-associated transcripts were captured from four adult ticks that were sequenced. RNA viruses of great diversity, especially those from the family *Bunyavirales*, were identified in relatively high abundances in the metatranscriptome. In contrast to targeted surveillance, metabarcoding by HTS is a cost-effective method to simultaneously detect diverse *B. burgdorferi sensu lato* and non-*B. burgdorferi sensu lato* *Borrelia* in ticks, which can inform risk assessment of TBDs in Canada.

Using HTS, our study revealed *Borrelia* sp. from the *B. burgdorferi sensu lato* complex at all three sampling sites and at a higher abundance in the dissected midgut and salivary gland tissues of some ticks compared to the same tissue of other ticks and whole-tick samples. We also detected the non-*B. burgdorferi sensu lato* relapsing-fever spirochete *B. miyamotoi* in higher abundance from the dissected tissues of a single tick and at lower frequency of occurrence than the *B. burgdorferi sensu lato*-associated ASV. Although *B. burgdorferi sensu stricto* was first recognized as the genospecies responsible for LD, several other pathogenic *B. burgdorferi sensu lato* genospecies that cause Lyme or other Lyme-like diseases are also transmitted by *I. scapularis*, including *B. mayonii*, *B. kurtenbachii*, *B. bissettiae*, and *B. andersonii* (9–14). Beyond the *B. burgdorferi sensu lato* complex, the causative agent of relapsing fever, *B. miyamotoi*, has been found in *Ixodes* species from other parts of North America (66). Our results confirm that both *B. miyamotoi* and *B. burgdorferi sensu lato* complex spirochetes can concentrate in the midgut and salivary glands of unfed ticks.

Few studies from Canada have used HTS, but Sperling et al. (60) found that *Borrelia*-positive ticks had a higher $\alpha$-diversity than *Borrelia*-negative ticks and a high variance in *Borrelia* sequence abundance (0 to 93%) from ticks identified as *Borrelia*-positive using quantitative PCR (qPCR) (60). Moreover, Sperling et al. (60) showed that qPCR is sensitive to low copy number for detecting *B. burgdorferi* from engorged adult female *I. scapularis* and that 16S rRNA sequencing could consistently confirm qPCR results for both negative and positive detection of *B. burgdorferi*. Therefore, the goal of our study was to explore variation in the dissected and whole-tick bacterial microbiomes in a LD hot spot, rather than to compare the sensitivity of an HTS protocol. Given that HTS sensitivity is proportional to sequence coverage and that HTS costs are rapidly declining, higher coverage HTS is quickly becoming a cost-effective alternative to qPCR, especially when one

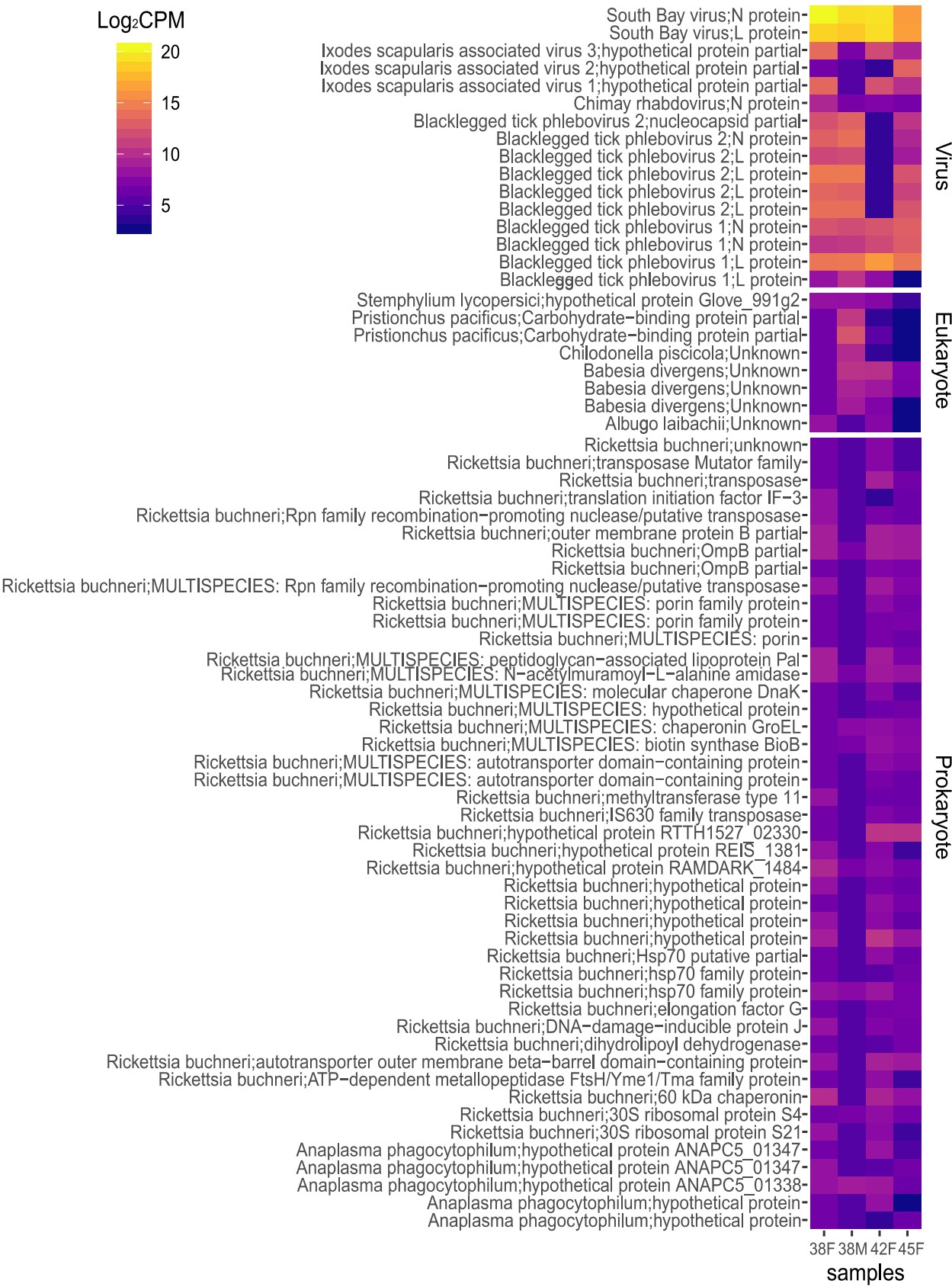

**FIG 4** Log$_2$ counts-per-million (CPM) expression of non-*Ixodes scapularis* assembled transcripts from transcriptome libraries of female (38F, 42F, and 45F) and male (38M) adult *I. scapularis*.

considers the cost of developing and testing multiplex primers, which cannot detect new and emerging pathogens.

In another study of the bacterial communities of adult *I. scapularis* from eastern and southern Ontario, Canada, Clow et al. (62) identified *Borrelia* sp. from 70% of the ticks tested with an average relative abundance of 0.01%. Similarly, we also identified a low abundance of *Borrelia* spp. from whole ticks, but the same sequences, when detected from some individual tick midgut and salivary glands, were found in much higher relative abundance comparatively. Our analysis highlights the value of metabarcoding the V4 16S rRNA from tick salivary glands and midguts, not only to identify potential pathogens and discriminate between *B. burgdorferi sensu lato* and non-*B. burgdorferi sensu lato Borrelia* but also to examine how the tick bacterial microbiome differs across tissues, which has implications for modeling risk associated of TBDs in Canada.

We found another potentially important human pathogen, *A. phagocytophilum*, at low frequency in this study, which was associated mainly with ticks collected from the QB site. Such findings are consistent with other PCR-based screenings that detected *A. phagocytophilum* infection frequencies from 0.4 to 8.9% in *I. scapularis* from Canada (67–69). Our work also demonstrates the value of metabarcoding by HTS in a phylogenetically informed analysis to resolve the strain-specific distribution of this potential pathogen. Specifically, we find strain Ap-var-1-CSF-21, which differs from the known pathogen strain Ap-ha-CSF-23 (58). Although the Ap-var-1-CSF-21 strain is not known to be pathogenic, human health risk assessments may be better informed by a more comprehensive sampling of the region to capture potentially virulent strains that may be present at lower frequencies within *I. scapularis* populations.

In addition to bacterial 16S sequences, we also detected *Babesia*-associated apicoplast sequences from the dissected salivary gland of two ticks collected from the QB site. Interestingly, the *Babesia*-associated sequence was coassociated with either *B. burgdorferi sensu lato* or *B. burgdorferi sensu lato* and *A. phagocytophilum*. While this *Babesia* sp. is not closely related to human-pathogenic *Babesia microti* based on BLAST hits, its pathogenic potential is unknown. It is reasonable to suspect a match to *Babesia odocoilei* because it has been detected in eastern Canada before (70–75), but we are not able to confirm this match since the apicoplast sequence is not available for *B. odocoilei*. A next step may include using targeted PCR, metabarcoding with *Eukaryote*-specific primers, or whole metagenomics to further classify this coassociated *Babesia* sp. detected from the QB site.

We went beyond previous 16S rRNA HTS studies in *I. scapularis* tissues (29, 65) to compare microbial communities across different tissues of individual ticks, from three sampling areas. Previous field studies by Zolnik et al. (65) and Ross et al. (29) pooled tissue from multiple individuals, which homogenizes tissue-specific community diversity. We overcame challenges of low DNA concentrations using a rigorous bioinformatic pipeline to filter, denoise, decontaminate, and remove chimeras. Additionally, we assigned ASVs to the species level, using LEfSe analysis, rather than clustering operational taxonomic units at the genus level (e.g., 97% similarity), which enabled the detection of more subtle differences in the microbiome. Applying these methods, we found considerable diversity among tissues, both within and among individuals. For example, bacterial communities of the salivary gland and midgut of individual 232 differed more from each other than any 2 whole-tick microbiomes, as shown in Fig. 3A. Quantifying the long-term temporal patterns associated with individual tick microbiome diversity from this emerging LD hot spot remains to be explored, but this will be crucial to estimating infection risk.

Despite the bacterial diversity we observe among ticks and tissue types, our methods appear to be validated by similarities to results of previous studies in *I. scapularis*. We detected three potential TBD-causing ASVs as discriminatory for tick tissues, including *A. phagocytophilum* (order *Rickettsiales*) and two different *Borrelia* spp. (order *Spirochaetales*), which is consistent with findings of Zolnik et al. (65), who found that *Anaplasma* and *Borrelia* spp. were relatively abundant in the salivary glands and midgut of adult female *I. scapularis* collected from New York state. Similarly, Ross et al.

(29) found that bacteria from the orders *Spirochaetales* and *Rickettsiales* were most likely to explain differences in the bacterial communities associated with tick internal viscera, compared with external washes, which is similar to our LEfSe results, in which *A. phagocytophilum* (order *Rickettsiales*) and two different *Borrelia*-associated ASVs (order *Spirochaetales*) discriminated between the communities associated with salivary glands and midgut tissues, compared with whole-tick samples. Furthermore, our finding that *Rickettsia buchneri* is the most common and abundant bacterial ASV in all organs of female ticks (Fig. 1) is consistent with results of the aforementioned studies (29, 65) and also findings of Al-Khafaji et al. (76), who confirmed rickettsial infection in *I. scapularis* salivary glands by confocal microscopy.

Salivary glands and midgut environments are important for the maintenance and transmission of tick-borne pathogens (77–80), in contrast to vertically transmitted endosymbionts like *R. buchneri* (24, 25). Our results suggest that horizontal transmission may impact the bacterial microbiomes in tick tissues differently than those in whole-tick samples, as tissues of individual ticks can host drastically different microbiota even though whole-tick microbiomes appear similar (Fig. 3). So, while micrometer dissections may be time-consuming, they can improve the detection of tick-borne pathogens and help model risks posed by *I. scapularis*.

In addition to requiring more time and expertise, HTS sequencing of microdissected tick tissues poses two additional challenges. First, the smaller amount of extracted DNA from tissue-specific sequencing is more prone to contamination (29). Second, the $\beta$-diversity measures of low biomass samples are more prone to inflation due to uneven sampling depth, whereas samples with relatively fewer sequences have greater uncertainty associated with rare ASVs (81–83). To address these issues, we applied a two-step approach to first screen contaminating ASVs from the sequencing data, followed by a rarefaction analysis to assess for sufficient capture of bacterial diversity (Fig. S4). Even after these corrective measures, we identified a greater number of unique ASVs associated with bacterial communities from dissected tissue than that from whole-body samples. This finding does not appear to be an artifact of small sample size or uncertainty associated with rare ASVs because $\alpha$-diversity in our lower biomass samples (i.e., dissected tissues) was not inflated relative to the whole-body samples in our rarefaction analysis (Fig. S4). The biological significance of higher among-tick diversity in tick tissue is worthy of future study.

As mentioned above, the most abundant ASV identified in our study was *Rickettsia* sp., which is a finding consistent with those of other studies of *I. scapularis* (29, 60–65). Moreover, we identified active RNA transcription of the *R. buchneri* biotin synthesis gene *bioB*, which is encoded on a plasmid along with all the genes for biotin synthesis (84). We also identified several *R. buchneri*-associated transcripts coding for known secreted factors identified previously by Al-Khafaji et al. (76), including transcripts for a peptidoglycan-associated lipoprotein and moonlighting proteins DnaK and GroL. In addition to these known proteins, we also identified *R. buchneri*-associated transcripts coding for various secretion systems, including porins, autotransporters, and autotransporter domain-containing proteins. Although *R. buchneri* is not a known human pathogen, our results imply that *R. buchneri* could play a role in tick feeding in ways that may interact with the immune system of the vertebrate host via transmission of rickettsial-secreted factors in tick saliva and gut contents.

Consistent with surveys of *I. scapularis* in other geographic regions (43, 85, 86), we detected *Bunyavirales*-like viruses, such as SBV, BLTPV-1, and BLTPV-2, as well as three unclassified ISAVs from the metatranscriptome of *I. scapularis*. These viruses are tick specific and are not known to be pathogenic to humans but could influence tick physiology and behavior. In contrast to the metatranscriptome reported by Cross et al. (43), in which they rarely observed BLTPV-1 and BTPV-2 co-occurring in the same *I. scapularis*, our study detected co-occurrence of both BTPV-1 and BTPV-2 in three out of four of the ticks analyzed, suggesting that both variants can coexist within the same host.

Among the female *I. scapularis* that we surveyed using 16S rRNA sequencing, *B. burgdorferi sensu lato*, non-*B. burgdorferi sensu lato Borrelia*, and *A. phagocytophilum*

were found in relatively high abundances, especially within the salivary glands and midgut. As a result, the entire bacterial microbiome was more variable among tissues of the same ticks than that among whole-tick samples. Beyond potentially pathogenic bacteria, 16S rRNA and metatranscriptome analysis also identified *Babesia* sp. of unknown virulence, while the metatranscriptome captured a diversity of *Bunyavirales*-like and other unclassified tick-specific viruses, some with relatively high expression. These results demonstrate the added value in combining both 16S rRNA and metatranscriptome sequencing for a more comprehensive view of the entire microbiome of *I. scapularis*, with a dynamic complex of pathogen and endosymbiotic microbes, including bacteria, viruses, and intracellular parasites.

## MATERIALS AND METHODS

**Tick collection and nucleic acid extraction.** Adult *I. scapularis* were collected at the Lemoine Point (LP) Conservation Area (44°13′, 56.228″N; 76°36′, 45.795″W) within the municipality of Kingston Ontario, Murphy's Point (MB) Provincial Park (44°46′, 54.7098′N; 76°14′, 30.1122′W), and Queen's University Biological Station (QB) (44°34′, 4.6524′N; 76°19′, 57.0966′W). Samples were collected from the QB site in 2016 and from all three of the sites in 2017. Questing adults were captured by flagging with 1-m² white flannel fabric attached to an aluminum bar along a transect of 25 m, checking for attached ticks at 5-m intervals. Ticks were removed by hand or with forceps, placed in 2-mL screwcap tubes containing 70% ethanol, and later stored at $-20$ or $-80°C$. Prior to extraction, ticks were submerged in a 1% bleach solution for 30 s and rinsed with lab-grade water. Different extraction methods were used for the DNA of internal tissue, DNA of whole ticks, and RNA of whole ticks, as follows.

We dissected 12 female ticks to remove internal tissue from the exoskeleton and to separate the salivary glands and midgut from the remaining internal viscera for separate extraction and sequencing. The midgut and salivary glands were isolated successfully from 10 ticks. Internal tissues were macerated separately in 50 $\mu$L of extraction buffer containing 1 mM EDTA, 25 mM NaCl, 10 mM Tris-HCl (pH 8.0), and 200 $\mu$g mL$^{-1}$ proteinase K (VWR). The DNA was purified using solid-phase reversible immobilization (SPRI) beads at 2.5$\times$ volume and resuspended in 20 $\mu$L laboratory-grade water.

In addition to the dissected ticks, we extracted DNA from 17 whole female ticks. Following sterilization as above, these ticks were dried at room temperature, frozen to $-80°C$, and pulverized in a Next Advance Bullet Blender Storm 24 device at 100 Hz in 2-mL tubes containing equal volumes of 0.2- and 0.5-mm low-binding ZrO beads (SPEX) for 3 to 4 min, refreezing every 1 min until no large visible fragments remained. Next, each sample was incubated with 500 $\mu$L of preheated cetyltrimethylammonium bromide (CTAB) buffer (100 mM Tris-HCl [pH 8.0], 25 mM EDTA, 1.5 M NaCl, 3% CTAB, 1% polyvinylpyrrolidone, and 1% $\beta$-mercaptoethanol) at 62°C for 30 min. To isolate the DNA, 500 $\mu$L of chloroform was added, the samples were centrifuged at 1,300 $\times$ *g* for 15 min, and then the supernatant was transferred to a new microcentrifuge tube. The DNA was precipitated in 1 mL prechilled 100% ethanol, mixed by inversion, and then incubated at $-20°C$ for 30 min. The DNA was pelleted by centrifugation at ~21,000 $\times$ *g* for 15 min, washed twice with 1 mL 75% ethanol, and resuspended in 15 $\mu$L laboratory-grade water.

We isolated RNA from one male and three female ticks using the same sterilizing and pulverizing steps as those for the whole-tick DNA protocol above but with 500 $\mu$L TRIzol reagent (Invitrogen) following the manufacturer's protocol and resuspending purified RNA in 15 $\mu$L lab-grade water.

**Library preparation.** The V4 region of the 16S rRNA gene was amplified from tick DNA using forward 515F (5′-GTGCCAGCMGCCGCGGTAA-3′) and reverse 806R (5′-GACTACHVGGGTWTCTAAT-3′) primers (87). The PCRs were completed in triplicate using a two-step approach, with each replicate, including a control PCR (template replaced with PCR-grade water). The first-step PCRs were completed in 25-$\mu$L volumes using ~2 to 3 ng of DNA template, 2 U Platinum *Taq* polymerase high-fidelity (Invitrogen), 0.5 $\mu$g · $\mu$L$^{-1}$ bovine serum albumin (New England BioLabs [NEB]), 3 mM MgSO$_4$, 200 $\mu$M dNTPs, and 0.4 $\mu$M each of 515F and 806R in 1$\times$ high-fidelity PCR buffer. Using a SimpliAmp thermal cycler (Applied Biosystem), the reaction mixtures were incubated at 94°C for 3 min; followed by 20 cycles of 94°C for 45 s, 53°C for 1 min, and 72°C for 1 min 30 s; and then a final elongation at 72°C for 10 min. These first-step products were purified using Sera-Mag magnetic SpeedBeads according to manufacturer directions, and 15 $\mu$L of this purified PCR product was used as the template for the second-step PCR for 12 cycles as above in 50-$\mu$L volumes. The second-step reactions were cycled 12 times using the above program with the 515F/806R phasing primers (88) to add the appropriate Illumina sequencing adapters, barcodes, and heterogeneity spacer components.

Following the second-step PCR, the products were then purified with SPRI beads as described above, and the concentration of the resultant DNA was quantified using the double-stranded DNA (dsDNA) high-sensitivity kit (DeNovix) on a DS-11 FX spectrophotometer/fluorometer (DeNovix) and pooled for equivalency. Twenty-four libraries failed to amplify and were excluded from the sequencing pool. Paired-end sequencing was performed at Genome Quebec using the MiSeq PE300 (2 $\times$ 300 bp) platform (Illumina V3 kit). A Sequence Read Archive listing of the sequencing data is available on GenBank under accession SRR17194087.

Tick RNA was assessed using the 2100 bioanalyzer (Agilent) to ensure sufficient quantity and quality before proceeding to rRNA depletion using the NEBNext rRNA depletion kit (human/mouse/rat). Next, the sequencing libraries were prepared using the NEBNext Ultra II directional RNA library prep kit for Illumina with NEBNext multiplex oligos for Illumina, and the resulting DNA concentrations were

determined with the NEBNext library quantification kit, which were all used according to NEB protocols. Finally, the cDNA libraries were sequenced at the Infectious Disease Sequencing Lab in the Kingston Health Sciences Centre using the MiSeq platform with a MiSeq reagent kit v2 (300 cycles) (Illumina MS-102-2002) producing 150-bp paired-end sequencing data. A Sequence Read Archive listing of the sequencing data is available on GenBank under accession SRR17194086.

**16S rRNA sequencing and analysis.** Raw sequence data were demultiplexed using cutadapt (v2.8) allowing up to 10% maximum error within the barcoded region (89). Libraries were demultiplexed based on barcode and heterogeneity spacer sequences and relabeled using the multiple move tool "mmv." The PCRs that failed to amplify during library prep were omitted from the data set prior to analysis. The remaining libraries were then processed and analyzed with a variety of packages for R (v4.1.0) (90) in RStudio (v1.2.5033) (91) using a reproducible analysis pipeline available online at https://github.com/damselflywingz/tick_microbiome (also see Data availability statement).

Forward and reverse libraries were filtered with default settings and trimmed using DADA2 for R (v1.20.0) (92). The unpaired forward and reverse libraries were each trimmed to a final length of 220 bp, after 5′ primer removal and trimming for poor-quality base calls in the 3′ regions. Next, a core sample inference denoising algorithm was applied to the pooled libraries (92). Following denoising, the forward and reverse pairs were merged and filtered for lengths between 250 and 265 bp (V4 region using 515F/806R is 254 bp). Identical sequences were collapsed into the longest representative sequence and screened for chimeras using the consensus method of *de novo* bimera removal.

Amplicon sequence variants (ASVs) were taxonomically assigned by implementing Ribosomal Database Project (RDP) (93) naive Bayesian classifier (94) using the DADA2-formatted training set 18 from v11.5 of the RDP (downloaded 26 August 2021). Species-level assignments were determined by comparing ASVs against the DADA2-formatted RDP species-assignment training set to identify unambiguous exact matches (95). The 16S V4 rRNA data set was converted into a phyloseq object and analyzed with "phyloseq" (v1.36.0) (96, 97).

Contaminant ASVs were removed with decontam (v1.12.0) by first applying the prevalence method on individual PCR batch replicate libraries and then applying the frequency method on the whole data set (98). During preliminary analysis, the libraries associated with one out of the three PCR batch replicates were found to contain 31 additional *Rickettsia*-associated ASVs that were not identified from the other two replicates. This problematic replicate was considered atypical compared with the other two PCR batch replicates and was subsequently removed from the analysis. Inconsistencies between low abundance ASVs across PCR replicate libraries were also removed by retaining only ASVs occurring in the remaining two replicate libraries unless only one library was available ($n = 7$). Libraries that were generated from any remaining replicate PCRs were combined *in silico*.

Any ASVs assigned at the phylum level within kingdom *Bacteria* were retained, except any ASVs assigned as "*Cyanobacteria/Chloroplast*" at the phylum level, which were removed. Only ASVs with counts greater than four were retained for further analysis. Any sample libraries with 7,500 or fewer ASVs were removed. Rarefaction curves were generated using "ranacapa" (v0.1.0) (99). The core bacterial community of *I. scapularis* was defined as the set of ASVs with ≥0.1% relative abundance in at least one library. Relative and total abundances of the core bacterial community were visualized with "ggplot2" (v3.2.1) (100).

Phylogenetic reconstructions of potential pathogenic *Borrelia*- and *A. phagocytophilum*-associated ASVs identified from the tick core microbiome were generated using 16S rRNA sequences available from NCBI 16S rRNA reference database (accession PRJNA33175; downloaded November 16, 2021) (101), and select sequences available through the NCBI GenBank website (101). The "phangorn" R package (v2.7.1) was used to generate optimized phylogenies for these ASVs (102). For each phylogeny, a maximum-likelihood tree was generated using a preoptimized distance-based method (i.e., rooted or unrooted), and the best substitution model was identified with phangorn. The branch confidence of each of the final phylogenetic reconstructions was tested with 1,000 bootstraps.

To assist with the taxonomic identification of known pathogens, we obtained 27 sequences of *Borrelia* and *Borreliella* 16S rRNA-associated sequences from the NCBI 16S rRNA reference database (101) and an additional sequence from *B. burgdorferi* isolate 15-0797 (accession MH781147.1) that did not contain any ambiguous bases. Potential *Borrelia* sequences from the above pipeline were aligned to these reference sequences using a staggered alignment with "DECIPHER" in R (v2.20.0) (103). Similarly, we used 11 *Anaplasma* and *Ehrlichia* sequences from the GenBank 16S rRNA reference database and two additional V4 16S rRNA sequences from strains of *A. phagocytophilum* that were detected previously in Ontario (58). Of these strains, the Ap-ha-CSF-23 (accession HG916766.1) strain is considered pathogenic, while Ap-var-1-CSF-21 (accession HG916767.1) is not known to be pathogenic (58).

Next, ASVs associated with the entire bacterial community were agglomerated at the lowest level of taxonomic identification. Linear discrimination analysis effect size (LEfSe) comparing 16S V4 rRNA libraries from (i) whole versus dissected tissue types across all sites and (ii) whole or dissected tissue types from each of the three sampling sites discriminated separately was undertaken using a linear discrimination analysis (LDA) score ($\log_{10}$) cutoff of 2.5 with "microbiomeMarker" (104). The V4 16S rRNA sequences from the entire bacterial community were next aligned with a staggered alignment using DECIPHER (2.14.0) using the profile-to-profile method (103). Using this alignment, a phylogenetic tree was constructed for the entire bacterial community with phangorn (v2.5.5) (102) using the neighbor-joining tree estimation function (105) with the generalized time-reversible model (106) (including an optimized proportion of variable size and gamma rate parameters) and stochastic tree rearrangement.

To evaluate patterns in the bacterial communities detected among whole or dissected individual *I. scapularis*, distance measures between the V4 16S rRNA libraries were calculated using phylogeny-aware weighted and unweighted UniFrac distances (107), ordinated by PCoA using phyloseq and "ape" (v5.4-1) (108) and labeled by sample location and tissue type. For unweighted UniFrac ordinations, convex hull

Gaussian ellipses were estimated using the Khachiyan algorithm (109) as implemented in the R package "ggforce" (v0.3.3) (110). A Venn diagram depicting common or unique ASVs detected in whole or dissected tick-associated libraries was generated with "VennDiagram" (v1.6.20) (111).

A permutation test of the difference in multivariate homogeneity of group dispersions (variances) between tissue and whole-tick libraries based on the unweighted UniFrac PCoA (see above) was implemented using vegan (v2.5-7). Nonparametric Wilcoxon signed-rank tests (112) were used to identify significant differences between mean unweighted UniFrac distances calculated from within and across libraries representing whole ticks and dissected tissues. To investigate how rare taxa influenced the results, the statistical analysis was repeated with the 500 most abundant ASVs (rather than the agglomerated data set described above) following normalization using total-sum scaling (counts from each ASV divided by total library size).

**Metatranscriptome analysis.** Paired-end sequence data were trimmed for quality and adapter removal of Illumina-specific adapter sequences (see Table S6 in the supplemental material) using Flexbar (v3.0.3) with trim-end mode "ANY," a minimum adapter overlap of 5, the maximum uncalled base of 1, and a minimum quality threshold of 20 (113). The metatranscriptome libraries from the four ticks were screened for mostly (92 to 96%) *I. scapularis* reads from the genome assembly (phased diploid assembly from ISE6 cell line) (accession GCF_00289285.2) (114) and PhiX (accession MN385565.1) using the splice-aware global aligner "BBMap" (v38.75) (115). The maximum indel size was increased to 200,000 against the host to accommodate large introns in the tick genome (116). The metatranscriptome was also filtered for residual amounts of prokaryotic rRNA and eukaryotic rRNA ($\sim$1 to 4%) using the "SortMeRNA" (v4.0.0) package (117). The remaining $\sim$2 to 3% resultant filtered sequence data were carried into the *de novo* assembly.

The metatranscriptome was assembled *de novo* using "Trinity" (v2.8.4) in default mode (118) and then clustered at 90% sequence identity threshold using "CD-HIT-EST" (v4.6.8) (119). Secondary clustering generated 6,868 assembled transcripts ($N_{50}$, 2,063 bp; $L_{50}$, 320 bp) but left a high proportion (82.8%) of unassembled sequences. Count tables were generated by aligning the filtered sequence data against the reference transcripts using "Salmon" (v0.14.1) (120). Successfully aligned sequences represented 29.1% of all paired-end reads. Prior to normalization, the raw count data were filtered for low abundance transcripts (i.e., fewer than five aligned reads in at least one sample). The count data were then normalized with the upper quartile scaling approach and transformed using the "voom" function implemented in "edgeR" (121) and "limma" (122, 123), respectively. The $\log_2$ counts-per-million (CPM) expression values were visualized with ggplot2 (100).

For taxonomic assignment to each of the reference transcripts, we used BLASTx (124) to search against the GenBank nonredundant (nr) protein database (101). Similarly, we also used BLASTn (124) to search against the GenBank nucleotide database (101) and the *R. buchneri* genome assembly REISMNv1 (accession GCF_000696365.1) (125). An E value retention threshold of $1e^{-5}$ was used for all BLAST-based searches. Taxonomy could be assigned to 2,206 (32.1%) reference transcripts by priority using matches from the *R. buchneri* genome with the lowest E value, over matches against any residual *I. scapularis* nonredundant proteins, and then followed by the lowest E value from either the nr protein or nucleotide reference databases. Taxonomic information was assigned based on the blast-generated taxonIDs using "TaxonKit" (v0.5.0) (126), and the final metatranscriptome was filtered for transcripts annotated as unwanted phyla, including *Arthropoda*, *Brachiopoda*, *Chordata*, *Cnidaria*, *Echinodermata*, *Mollusca*, *Priapulida*, *Streptophyta*, and *Tardigrada*.

**Data availability.** Accession numbers SRR17194086 and SRR17194087 have been deposited into the GenBank Sequence Read Archive (SRA) representing the raw sequence data analyzed in this study. We have made these data available without restriction.

We have also made available, without restriction, the details related to the programs and algorithms used to analyze these sequencing data, including suitable documentation regarding their use, as a reproducible analysis pipeline through GitHub. Please see https://github.com/damselflywingz/tick_microbiome (archived version https://doi.org/10.5061/dryad.fqz612jw9) and the Materials and Methods for the relevant details. This analysis pipeline also includes the computer code created to interpret data and generate the results presented in this study.

## SUPPLEMENTAL MATERIAL

Supplemental material is available online only.

**SUPPLEMENTAL FILE 1**, PDF file, 1.2 MB.

## ACKNOWLEDGMENTS

We acknowledge C. Brdar and E. Barkley of Ontario Parks and S. Knapton of the Cataraqui Region Conservation Authority for authorizing the collection of tick samples used in this study. We are grateful to K. Ding and A. Siew of Queen's University Biology Department, who performed the tick dissections and extractions. We are also grateful to Z. Sun and G. McLeod of Queen's University Biology Department, who carried out the library preparations. We appreciate S. Afsharnezhad and L. Wisteard for debugging the bioinformatic pipeline. We acknowledge that S. Patel provided feedback on an early version of the manuscript.

A.R.P., Bioinformatics, formal analysis, data curation, writing – original draft, visualization; D.H., Bioinformatics, validation, writing – review & editing; S.C.L., Conceptualization, methodology, investigation, writing – review & editing; R.I.C., Conceptualization, methodology, validation, investigation, resources, writing – review & editing, supervision, project administration, funding acquisition.

This work was funded by New Frontiers in Research Fund, Exploration Grant and Ontario Government Early Researcher Award, and funding from Queen's University to R.I.C. The funders had no role in study design, data collection and interpretation, or the decision to submit the work for publication.

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
