## [Reviewer comments · Microbiology Spectrum]

Microbiology Spectrum

Multi-omics reveals symbionts, pathogens and tissue-specific microbiome of blacklegged ticks (*Ixodes scapularis*) from a Lyme disease hotspot in southeastern Ontario, Canada.

Amber Paulson, Stephen Lougheed, David Huang, and Robert Colautti

Corresponding Author(s): Amber Paulson, Queen's University

Review Timeline:

Submission Date:

April 17, 2023

Accepted:

April 24, 2023

Editor: Catherine Brissette

Reviewer(s): The reviewers have opted to remain anonymous.

Transaction Report:

DOI: <https://doi.org/10.1128/spectrum.01404-23>

April 24, 2023

Dr. Amber R Paulson
Queen's University
Biology
Kingston, Ontario
Canada

Re: Spectrum01404-23 (Multi-omics reveals symbionts, pathogens and tissue-specific microbiome of blacklegged ticks (*Ixodes scapularis*) from a Lyme disease hotspot in southeastern Ontario, Canada.)

Dear Dr. Amber R Paulson:

As stated by the single reviewer on the previous version, "The manuscript is well-written and the methods are sound." The authors addressed the previous critique, and the novelty of looking at tissues from individuals rather than communities in an area of recent range expansion is novel.

Your manuscript has been accepted, and I am forwarding it to the ASM Journals Department for publication. You will be notified when your proofs are ready to be viewed.

Sincerely,

Catherine Brissette
Editor, Microbiology Spectrum
